

# Predicting the unemployment rate and energy poverty levels in selected European Union countries using an ARIMA-ARNN model

Claudiu Ionut Popirlan, Irina-Valentina Tudor and Cristina Popirlan

Department of Computer Science, University of Craiova, Craiova, Dolj, Romania

## ABSTRACT

This article analyzes the correlation between energy poverty percentage and unemployment rate for four European countries, Bulgaria, Hungary, Romania and Slovakia, comparing the results with the European average. The time series extracted from the datasets were imported in a hybrid model, namely ARIMA-ARNN, generating predictions for the two variables in order to analyze their interconnectivity. The results obtained from the hybrid model suggest that unemployment rate and energy poverty percentage have comparable tendencies, being strongly correlated. The forecasts suggest that this correlation will be maintained in the future unless appropriate governmental policies are implemented in order to lower the impact of other aspects on energy poverty.

## INTRODUCTION

A country's economic development is impacted by the energy sector, the stability in energy prices having an important role in expanding the economic growth. *Wang et al. (2019)* suggest that the instability in energy prices, particularly their increase, encourages the inflation leading to wage rates reduction and this generates unemployment.

When considering general economic growth, energy usage is of outstanding importance due to various forms of energy conversion and the important role this has in the production process. Depending on the source of energy, it could have different impacts on various sectors of the economy and also on the surrounding environment. A lot of energy research has concentrated on the disponibility of renewable energy alternatives in order to alleviate the effects of climate change (*Apergis & Payne, 2010*). Results in this area show that there is a connection between economic growth and renewable energy expenditure that seem to be interdependent (*Menyah & Wolde-Rufael, 2010*).

According to the *World Bank (2022a)*, there are approximately one billion people, around 9% of the world population, who did not have access to electricity in 2020. Concerning this, all European households have access to the electric grid; however, a

Corresponding author
Claudiu Ionut Popirlan,
popirlan@inf.ucv.ro

challenge for governmental policies in the EU is to find efficient solutions to turn existing energetic systems into sustainable renewable energy systems.

Millions of people around the world are affected by energy poverty; in developing countries the main issue is clean energy accessibility, while in developed countries the major challenge is high energy prices that prevent satisfying appropriate basic energy needs. Due to various implications that energy poverty has on the economic, social and political environment, European countries try to prioritize managing this issue.

Reviewing the literature, we observe that researchers tried to define, as closely as possible, the notion of energy poverty. A first interpretation of this concept given by *Lewis (1982)* was referred to as being able to keep the household adequately warm, while now, as the *European Commission (2022)* considers, it comprises all basic energy needs of a household (maintaining the house cool during summertime is also included in the basic needs).

Several indices and methods have been used over time to address energy poverty measurement, each of them analyzing different aspects of this phenomenon, such as: energy poverty generated by low-income or unemployment, or the amount of money used for energy costs. For example, *Papada & Kaliampakos (2020)* say that the 10% index measures the percentage of income spent on household maintenance.

A well-known method to measure energy poverty is the FGT (Foster, Greer and Thorbecke) indices described by *Foster, Greer & Thorbecke (1984)*, which presents an unifying framework based on a parameter whose value changes according to the analyzed situation. The first FGT measurement is just a headcount index of the households below the poverty line, providing the percentage of population that use energy below their basic needs.

As described in *Mashhoodi, Stead & van Timmeren (2019)*, although energy poverty is a predominant issue, many countries do not separate it from general poverty, hence the limited policies dedicated to this specific matter. Most governmental policies do not focus on reducing long-term energy poverty but tend to designate resources for assisting energy vulnerable households. In some of the considered countries the governments have implemented programs that offer energy poverty assistance. However these are insufficient unless governments take appropriate measures in order to reduce energy poverty in households.

Aside from constant energy poverty issues, the countries could face other energy related challenges: energy production may contaminate the environment or energy supply may be at risk of disruptions. Renewable green energy may be a legitimate solution to solve the addressed challenges.

*Karpinska & Smiech (2020)* found that around 23.57% of the Central and Eastern European population suffers from hidden energy poverty, many of them living in less urbanized areas or in one-person households. Hidden energy poverty refers to the situation where the amount of money left after total housing expenditures is below a minimum level established for each country, according to the standard relative poverty boundary from *Eurostat (2022a)*.

Some individuals living in energy poverty households cannot afford to keep home adequately warm due to their low income and employment status. Economic growth is influenced by investments that are discouraged by a rise in the unemployment rate, consequently leading to poverty, particularly to energy poverty.

There is a well defined relationship between energy consumption and the energy price, the energy market suffers insignificant adjustments when the price goes through substantial changes. Considering the situation when the energy input decreases and all the other economic aspects remain constant (*Bretschger, 2015*), we observed that energy prices will increase leading to energy poverty of households.

In 2021, 6.9% of the European population is considered to be in energy poverty, unable to keep their home adequately warm according to *Eurostat (2022b)*. We observed that in the considered developing countries this percentage is greater than the European average value, Romania with 10.1% and Bulgaria at the bottom of the European Union (EU27) list, with 23.7%, while the developed countries have a lower ratio Hungary with 5.4% and Slovakia with 5.8%.

High energy prices lead to labor productivity reduction, unemployment increase and put pressure on country policies to develop alternative energy saving solutions for industrial production. An alternative policy that could aid lowering the unemployment rate could be labor or capital redistribution among productivity sectors. A decrease in energy prices also decreases the country's currency. The natural conclusion, presented by *Arshad, Zakaria & Junyang (2016)*, is that energy price is directly proportional to economic growth and indirectly proportional to exchange rate and unemployment.

Reviewing the field literature, we can select the main energy-related aspects analyzed over time:

- Energy consumption and energy prices are mutually dependent;
- Unemployment impacts households income;
- Low income due to unemployment leads to energy poverty;
- Energy poverty is strongly influenced by income;
- Energy consumption and energy prices impact the energy poverty percentage of population;
- Hidden energy poverty is one of the main topic in energy poverty studies;
- Constant energy price growth determines the transition towards renewable energy.

*Zeren & Akkus (2020)* explained that there is a cyclic relationship between energy consumption, energy price and income which is highly dependent on the employment status. Considering this, the high energy prices lead to energy substitution usage such as renewable solutions.

Energy prices constitute a large percentage of the total housing expenditures, and besides that, individuals exposed to hidden energy poverty tend to cut down expenses with lightning, heating or cooling depending on the season, washing and cooking, because all of these are energy-dependent.

The situation when an employable person (excluding people with disabilities, retired persons or students) is unable to find a job, but is actively searching for one, is referred to

as unemployment. The unemployment rate is one of the main indicators of the economy. A large number of unemployed individuals leads to a low economic productivity and, during the unemployment period, people should not diminish their basic consumption needs (*Soylu, Cakmak & Okur, 2018*).

A consistent high unemployment rate means the economy is in difficulty and that could generate revisions in the socio-political environment. On the other hand, a low unemployment rate means that the economy is production-based, gradually leading to improved living standards and increased income.

Unemployment can be voluntary, caused by leaving the current job and trying to find another, and involuntary, caused by recession, technological advancement, job outsourcing (*Petrosky-Nadeau & Valletta, 2020*).

Latest economic research (*Schmidpeter & Winter-Ebmer, 2021*) suggests three types of unemployment: frictional (refers to voluntary passage from one job to a better one), structural (refers to unemployment caused by technological advancement, insufficient working skills or companies moving their business to another location) and cyclical (refers to unemployment caused by changes in business periodicity).

The unemployment rate in the EU27 at the end of September 2022 was at 6%, while the four countries selected for this study are all below the European average as follows: Bulgaria 4.6%, Hungary 3.7%, Romania 5.2% and Slovakia 5.9% according to *Destatis (2022)*.

Over time, various hybrid models have been developed and used for different prediction problems in studies for the financial and socio-economic environment, medical field (*de Araujo Morais & da Silva Gomes, 2022*) and other areas.

ARIMA model is suitable for linear forecasting while ARNNs are more convenient for nonlinear predictions, but considering the fact that real-world problems are complex, using both models into a combined hybrid model seems to be the optimal approach and thus we considered this solution for making predictions in our study.

In a previous study, we predicted the unemployment rate for several European countries obtaining results that were mostly validated in the last year, driving us to continue investigating the unemployment rate prediction in correlation with other aspects for the above-mentioned countries (*Popirlan et al., 2021*).

In this article we study the correlation between the percentage of population in energy poverty and the unemployment rate, using available European datasets. We selected two developing countries, Romania and Bulgaria, and two developed countries, Hungary and Slovakia, and we analyzed the connection between energy consumption, energy poverty and unemployment rate for each of them, in comparison with each other and with the average values of the European Union.

Then, for the selected European countries we explored the forecasting potential of the ARIMA-ARNN hybrid model, generating one year ahead predictions for the unemployment rate and energy poverty percentages. In order to obtain the above-mentioned predictions, we analyzed the monthly available data for unemployment rate from 2000 to 2022. For predicting the percentage of energy poor population, the available statistics start from 2005 until 2021.

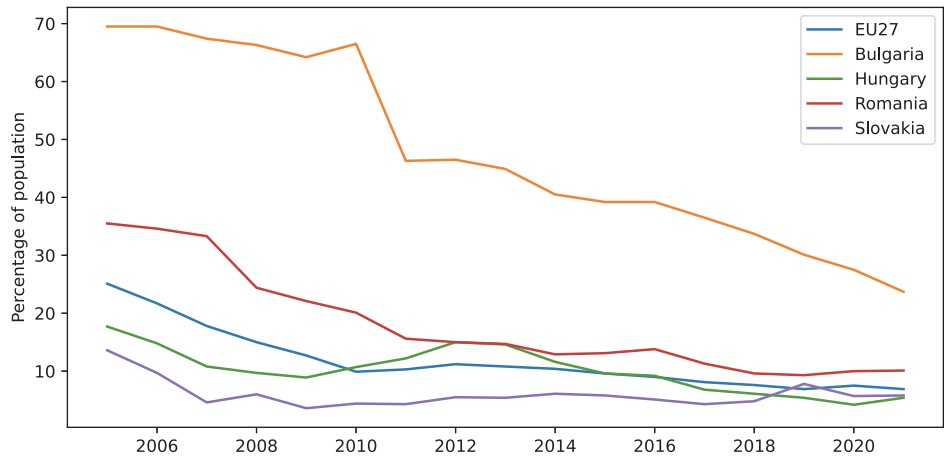

Figure 1  **Trends of the energy poverty percentages.**

## METHODS AND RESULTS

### Data analysis

For this study we used the available data on *Eurostat (2022b)* extracting all the information for Sustainable Development Goal 7—Affordable and clean energy from the *United Nations Department of Economic and Social Affairs (2022)*. In addition to an empirical data analysis, we also generated a graphical representation (Fig. 1) of the extracted values in order to better understand the trend energy poverty had over time in the considered countries and in comparison with the European average. As we can observe in the graphical image, the percentage of energy poverty population is below the European average in the developed countries (Hungary and Slovakia), while in the developing countries(Romania and Bulgaria) this percentage varies above the average value. It's easy to see that for Bulgaria the energy poverty percentage is much higher than in EU27 and the other countries, even if in the recent years it had a continuous descending trend. For Hungary, Romania and Slovakia we notice that the percentage of population in energy poverty had a slightly ascending tendency in 2020 and 2021, most likely due to the COVID19 pandemic.

We estimate these percentages will continue to increase in the next period due to the actual political and economic situation in Europe.

Governments try to decrease the percentage of population that cannot afford to keep their home warm by introducing policies that facilitate renewable energy access and diminish the ascending trend of energy prices. Individuals and institutions started reducing their energy consumption in order to avoid the situation when they cannot afford the energy costs so as not to become energy poor (*Residential Energy Efficiency (REE) Observatory in Central and Eastern Europe (CEE), 2022*).

In our analysis we used the unemployment rate datasets for the last 23 years acquired from the *World Bank (2022b)*. For a better visualisation of the data, we plotted the unemployment rate for the selected countries only for the last 17 years as shown in Fig. 2. We notice that

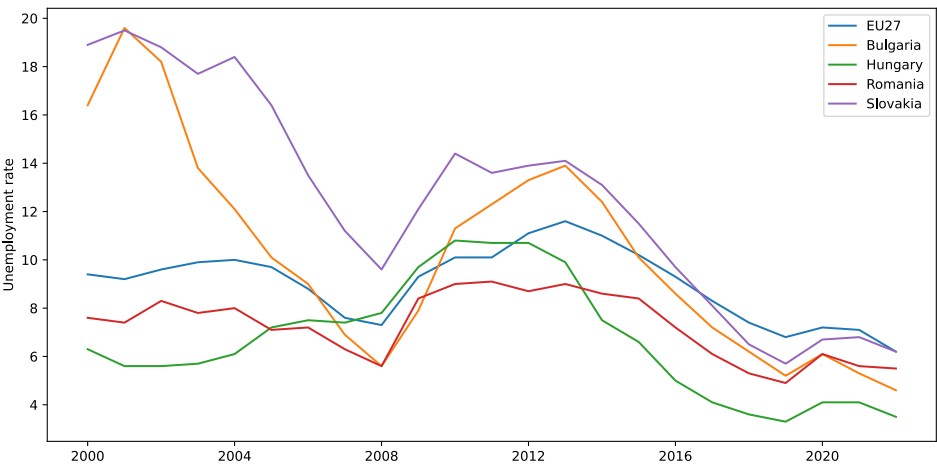

**Figure 2  Graphical representation of the unemployment rate.**

**Table 1  Pearson correlation for unemployment rate (UR) and energy poverty (EP).**

| Country | EP & UR | EP & EU27 EP | UR & EU27 UR |
|---|---|---|---|
| Bulgaria | 0.616 | 0.822 | 0.957 |
| Hungary | 0.887 | 0.735 | 0.716 |
| Romania | 0.748 | 0.958 | 0.908 |
| Slovakia | 0.790 | 0.691 | 0.807 |

the unemployment rate in the middle of 2022 in EU27 and in the considered countries as well, has nearly reached the same values as before the COVID19 pandemic presented by *Ahmad et al. (2023)* which produced a massive increase of unemployment in the summer of 2020 worldwide.

Further, for a better data analysis, we computed the Pearson correlation between the unemployment rate and the population unable to keep their home adequately warm by poverty status datasets, and for every country we compared each variable with the EU27 value. We can conclude that there is a positive correlation between these two variables. The results are summarized in Table 1 and all the values have correlations significant at the 0.01 level (two-tailed).

In the European Union the direction for the unemployment rate is given by the developing countries, their values having the most influence on the European unemployment average. The same goes for the percentages of the energy poverty population. The developed countries have values close to the European average in both cases, for unemployment rate and for energy poverty. The best correlation coefficient between the two variables is observed for Hungary dataset, mainly because the government policies manage to set a balance on the labor market, significantly reducing the untaxed work.

The unemployment rate serves as a key economic indicator, spanning various areas of interest, applied in investment policies because it is strongly correlated with the business rhythm and it's dominant in monetary management. Forecasting the unemployment rate influences the decisions made in the socio-economic environment and we consider energy poverty as being one of the variables impacted by unemployment.

## ARIMA-ARNN model

Over time, the AutoRegressive Integrated Moving Average (ARIMA) model has been extensively used for predicting stochastic time series with applications in economy, unemployment rate, inflation, stock market, electricity price, daily wind speed, traffic congestion, crime prediction, water treatment, energy consumption, air pollutants, daily solar energy production or COVID-19 pandemic (*Parreno, 2022*; *Arun Kumar et al., 2022*; *Jinnah, Sahib & Ibrahim, 2022*).

Analyzing different forecast examples in various fields, it can be observed that ARIMA obtains promising results and over time the predicted values proved to be comparable to the actual measured values. The ARIMA model relies on the linearity of time series and in many research studies the values are assumed to be linearly correlated which works fine for simple scenarios. However, the real world is more complex and there are many situations where time series also have a nonlinear component that needs to be addressed as well. In order to obtain satisfactory results, complex problems are more likely to be simulated using both the linear and nonlinear components of the time series values leading to a hybridisation of the considered model.

To capture the complex real world situations, a flexible tool has been used, namely artifical neural networks (ANN) described by *Tealab (2018)*, which, in comparison with ARIMA, proved to be more effective only in some particular cases. An important feature of ANN models is their capability to design the time series nonlinear component by using multiple layers for interactions between the nonlinear neurons.

When modeling a time series with ARIMA, it was noticed that the model cannot seize the nonlinear side of these values; thus, it is necessary to use a different modeling technique that deals with the nonlinearity, such as the multilayer ANN model. Due to the heterogeneity of the time series components and their underlying patterns, a hybrid approach is suitable for obtaining accurate forecasting, reducing disparities of forecast errors of the involved models as proved by *Buyuksahin & Ertekin (2019)*.

Since the ANN model is complex and it's hard to find an optimal network architecture, a less complex and easier to describe model is recently preferred by researchers, being easily fitted to time series that uses its lagged values as inputs. This AutoRegressive Neural Network (ARNN) model described by *Farhi & Yasir (2022)* represents a feed-forward neural network with a single hidden layer.

*Chakraborty et al. (2021)* proposed an ARIMA-ARNN hybrid model and theoretically proved its asymptotic stationarity leading to accurate forecasting of wide time span series without increasing variance over time. They also show an example of applying the hybrid model on unemployment rate forecasting for several countries.

The residual errors generated by applying the ARIMA model on the time series after transforming it into a stationary one are then modeled using the nonlinear ARNN model.

ARIMA is a linear model with three parameters $ARIMA(p,d,q)$, where $p$ is the order of the autoregression model, $q$ is the order of the moving average model and $d$ is the level of differencing for converting the nonstationary time series into stationary. Based on the time series complexity, the differencing process may need to be repeated.

The ARIMA model is composed of two submodels, the AR model and the MA model, described mathematically as follows:

$$Y(AR)_t = \alpha + \beta_1 Y(AR)_{t-1} + \beta_2 Y(AR)_{t-2} + \cdots + \beta_p Y(AR)_{t-p} \tag{1}$$

where $Y(AR)_t$ are the lags of the series, $\beta$ are the lags coefficients estimated by the model and $\alpha$ is the model's intercept term.

$$Y(MA)_t = \epsilon_t + \theta_1 \epsilon_{t-1} + \theta_2 \epsilon_{t-2} + \cdots + \theta_q \epsilon_{t-q} \tag{2}$$

where $Y(MA)_t$ depends only on the lagged forecast errors, $\theta$ are the parameters of the model and $\epsilon$ are unrelated error terms.

The mathematical expression of ARIMA model, as a combination of AR and MA models, is:

$$Y_t = \alpha + \sum_{i=1}^{p} \beta_i Y_{t-i} + \epsilon_t - \sum_{j=1}^{q} \theta_j \epsilon_{t-j} \tag{3}$$

where $Y_t$ and $\epsilon_t$ represent the time series and the random error at time t, $\beta$ and $\theta$ are the coefficients of the model.

For estimating the $p$ and $q$ parameters of the AR and MA models we used plots for the autocorrelation function (ACF) and the partial autocorrelation function (PACF). Bayesian information criterion (BIC) and Akaike information criterion (AIC) presented by *Vrieze (2012)* were used to determine the best ARIMA model.

ARNN is a nonlinear modified neural network with two parameters $ARNN(p,k)$, where $p$ is the number of lagged inputs (from the AR part of the model) and $k$ is the number of neurons from the hidden layer of the model. If the time series values are non-seasonal then the value of k is $[\frac{p+1}{2}]$. BIC is also used as the main test for choosing the best ARNN model. The mathematical expression of the ARNN model is:

$$X_t = \phi_0 \{ w_{c_0} + \sum_k w_{k_0} \phi_k (w_{c_k} + \sum_{i=1}^{p} w_{i_k} X_{t-j_i}) \} \tag{4}$$

where $X_t$ is the time series at the moment $t$ computed using the previous observations, $w_c$ are the associated weights and $\phi$ represents the activation function.

The hybrid model considered for this study is ARIMA+ARNN (as shown in Fig. 3):

$$Z_t = Y_t + X_t \tag{5}$$

where $Y_t$ is the linear component obtained using the ARIMA model and $X_t$ is the nonlinear component computed with the ARNN model.

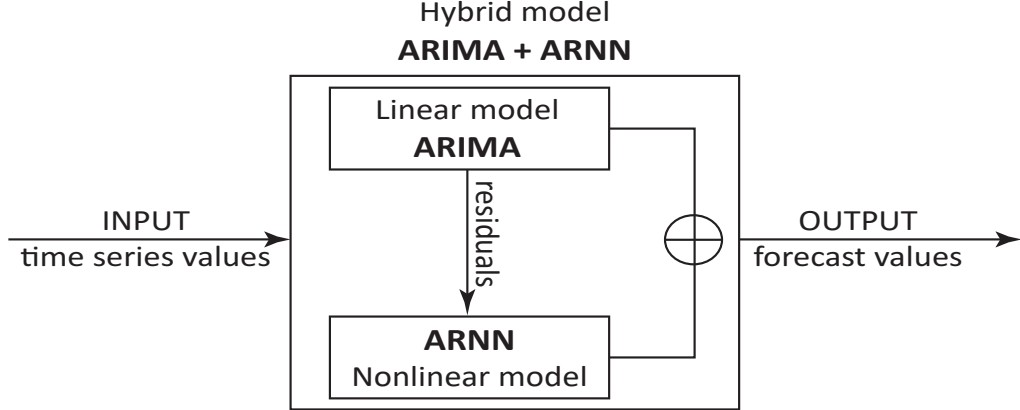

**Figure 3  AARIMA+ARNN hybrid model architecture.**

The residuals autocorrelations that could not be modeled by ARIMA are introduced in ARNN that models them leading to improvement of the original forecasts. The method used for forecasting using this hybrid model can be summarized in the following steps:

1. Determine the most appropriate parameters p, d, q for ARIMA, using the ACF and PACF plots and AIC and BIC coefficients;
2. Apply previously determined ARIMA(p,d,q) model for the training data set;
3. Determine the forecast values and residual errors from the ARIMA implementation;
4. Import residuals in the ARNN(p,k) model;
5. Compute the forecast values of the ARNN model;
6. Obtain the final forecast by combining the forecasts of ARIMA and ARNN.

## Results

The ARIMA-ARNN hybrid model is used in our study for predicting the unemployment rate and energy poverty percentage thus analyzing the connection between the two variables.

Time series associated with the unemployment rate of a country is nonlinear and nonstationary; therefore, a hybrid prediction model would be the most appropriate for dealing with these variations in the data. The hybrid model has been theoretically demonstrated and it seems to be asymptotic stationary when forecasting over large periods of time.

We tested the stationarity of every dataset using the augmented Dickey-Fuller (ADF) test (*Dickey & Fuller, 1979*; *Cheung & Lai, 1995*). Almost all of the series were nonstationary when first applying the ADF test, but, after applying first order differencing, all the series failed to reject the null hypothesis of stationarity, thus giving the value 1 for the differencing parameter of the ARIMA models.

The unemployment rate datasets, distributed over almost 23 years, between 2000 and 2022, are divided into training sets and test sets (12 months).

First, we applied the model on the unemployment rate for the considered datasets and obtained the following results:

- For EU27, ARIMA(5, 1, 5) was the best obtained option, having AIC $-167.075$. We extracted the residuals from ARIMA and trained them further, obtaining the ARNN(24,12) model. Then we extracted the predictions from the ARIMA+ARNN model, comparing them with the test dataset and obtaining the following accuracy: RMSE $= 0.144$, MAE $= 0.128$.
- For Bulgaria, ARIMA(5,1,4) was the best option, having AIC 53.277. The residuals extracted from ARIMA were trained further, obtaining the ARNN(12,6) model. The ARIMA+ARNN model was applied, predictions were computed and then they were compared with the test dataset: RMSE $=0.118$, MAE $=0.094$.
- For Hungary, we obtained AIC 230.241 for ARIMA(5,1,5) and ARNN(15,8) with an average of 20 networks. Forecasts were extracted from the ARIMA+ARNN model, obtaining an accuracy of RMSE $= 0.310$, MAE $= 0,275$.
- For the Romanian dataset we obtained ARIMA(4,1,3) with an AIC $= 228.33$. Its residuals were trained with the ARNN(12,6) model and then we applied the hybrid ARIMA+ARNN model, with the following accuracy: RMSE $= 0.257$, MAE $= 0.214$.
- For Slovakia, the best ARIMA was for $p = 2$, $d = 1$, $q = 3$ with AIC $= 40.112$ and the best ARNN model for residuals training was ARNN(24,12). Then, the accuracy of RMSE $= 0.082$ and MAE $= 0.074$ were obtained by computing the predictions with the ARIMA+ARNN model.

For every residual series collected from the ARIMA models we used the Ljung–Box portmanteau test (*Ljung, 1986*) with five lags, in order to examine if the residuals are independently distributed. The *p*-values we obtained are greater than 0.05, therefore confirming the null hypothesis: EU27 *p*-value $= 0.890$, Bulgaria *p*-value $= 0.409$, Hungary *p*-value $= 0.942$, Romania *p*-value $= 0.876$, Slovakia *p*-value $= 0.758$.

Based on the results described above and considering the accuracy of the hybrid model we can use the model to obtain forecasts for the following 12 months. In Fig. 4, for every country and also for EU27, we represented with blue 24 months of actual unemployment rate values, the next 12 months are illustrated with blue for the actual values (test dataset) and with red for the predicted values obtained from the hybrid model, while the forecast values for unemployment rate are displayed in red for the final 12 months in the graphical representation.

The energy poverty datasets were split into training sets (2005–2020) and testing sets for 2021 (12 months).

Secondly, we applied the model on the energy poverty percentages for the collected datasets and obtained the following results:

- For EU27, ARIMA(4, 1, 4) was the best option having AIC 277.494. We extracted the residuals from ARIMA and trained them further, obtaining the ARNN(12,6) model with an average of 20 networks. Then, we extracted the predictions from the ARIMA+ARNN model, comparing them with the test dataset and obtaining the following accuracy: RMSE $= 0.267$, MAE $= 0.194$.

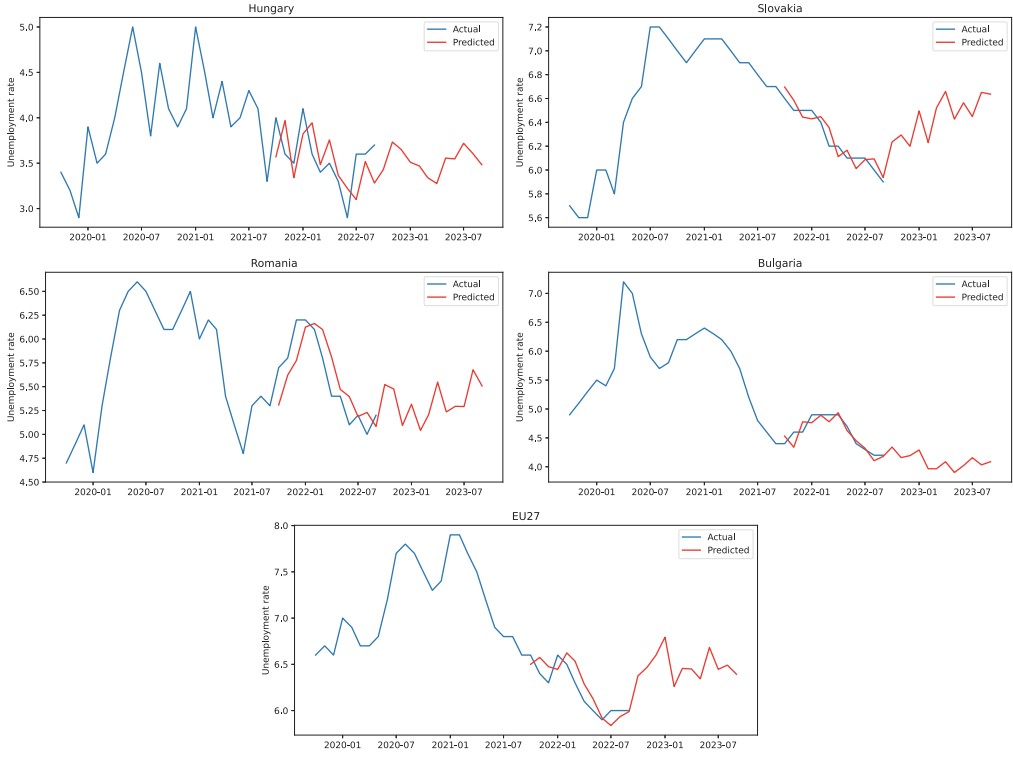

**Figure 4  ARIMA+ARNN forecasts for the unemployment rate.**

- For Bulgaria, an AIC of 694.454 was obtained for ARIMA(2, 1, 1) and, further, ARNN(5,3) was applied on the ARIMA residuals. An accuracy of RMSE = 0.216, MAE = 0.316 was retrieved for the hybrid ARIMA+ARNN model's predictions.

- For Hungary, ARIMA(3, 1, 2) was fitted on the training dataset with an AIC = 264.814. The derived residuals were then processed with the ARNN(12,6) model with an average of 20 networks, each of which is a 12-6-1 network with 85 weights. The final predictions from the hybrid ARIMA+ARNN model were compared with the test dataset obtaining: RMSE = 0.299, MAE = 0.254.

- For Romania, we obtained ARIMA(1,1,2) having AIC = 401.665 and ARNN(3,2) models. The predictions derived from both ARIMA and ARNN models are combined in order to determine the final test forecasts that led to the following accuracy values: RMSE = 0.216, MAE = 0.171.

- For Slovakia, forecasts were extracted from the hybrid ARIMA+ARNN model, based on ARIMA(1,1,1) with AIC 290.410 and ARNN(1,1) models, for which RMSE = 0.171 and MAE = 0.162 were computed.

The Ljung–Box test was again applied for the residual data obtained from ARIMA models, demonstrating the model performance: EU27 $p$-value = 0.642, Bulgaria $p$-value = 0.999, Hungary $p$-value = 0.907, Romania $p$-value = 0.706, Slovakia $p$-value = 0.892.

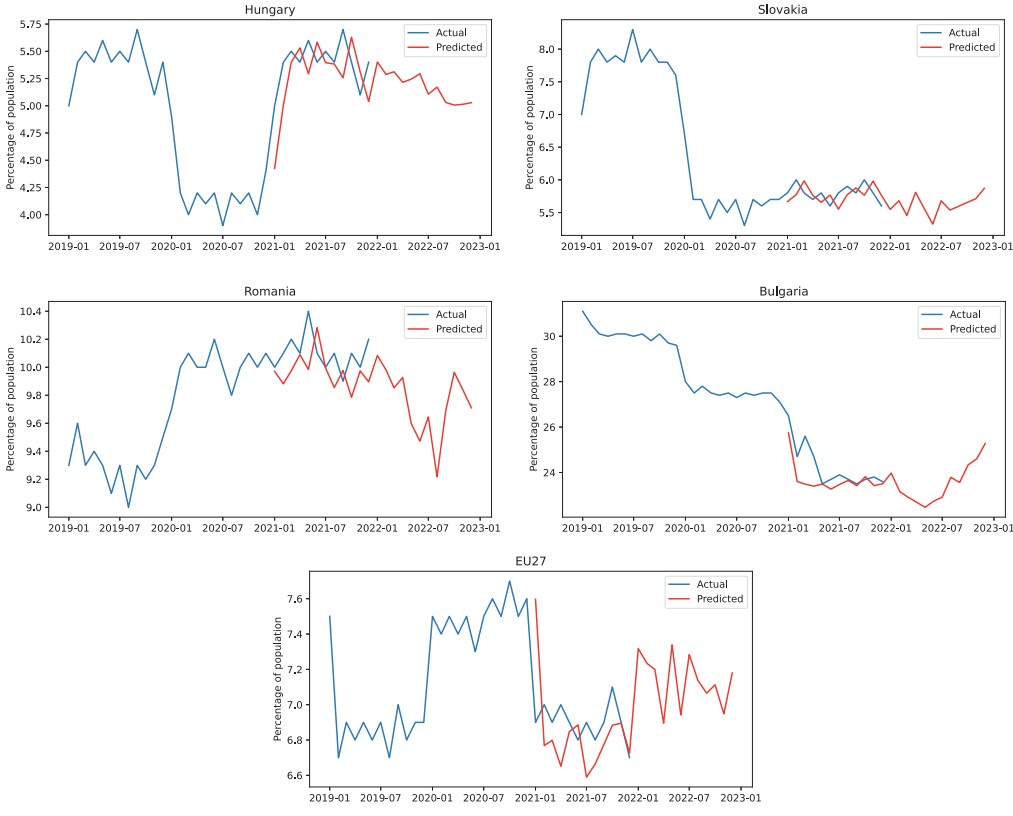

**Figure 5  ARIMA+ARNN predictions for the energy poverty percentage of the population.**

In Fig. 5, we plotted with blue lines the actual values of percentage of population in energy poverty for the last 24 months of the training dataset plus 12 months of the testing set compared to the testing prediction, followed by future 12 months forecasts. All the predictions were represented in our graphical interpretation using red lines.

## Discussion

Analyzing the forecasts we obtained from the hybrid model, we observe that, for unemployment, Bulgaria and Hungary are below the EU average, Romania is slightly below the average, while Slovakia has a similar trend compared to the European average.

Energy poverty forecasts present obvious differences between the developed and the developing countries. The two developed countries, Slovakia and Hungary, have lower energy poverty percentages than in EU27, while the developing countries, Romania and Bulgaria, have much higher percentages of energy poverty.

Bulgaria has a low unemployment rate, but has the highest energy poverty percentage among the four analyzed countries, because they have a low income, their houses have poor energetic efficiency although energy prices are lower than other countries in EU. The Bulgarian policies concerning energy poverty were applied mostly on dwellings rehabilitation and financial aid for the vulnerable population.

Romania is above the European average both on the income percentage spent on energy bills and also on the percentage of households that self-impose restrictions in order to decrease their energy expenditure. Energy prices in this country are slightly below the European mean, but the average income is much lower compared to the developed countries. Governmental policies are mainly focused on providing financial aid for households with low income. These regulations are disadvantageous for hidden energy poverty individuals, pushing them to restrict their consumption thus reducing their living confort. Houses rehabilitation and renovation policies are developed for the general population and do not apply specifically to energy poverty.

Compared to the developing countries, the developed ones tend to have a lower level of energy poverty. One of the main reasons is that they have a higher average income and so the hidden poverty has a reduced rate.

In Hungary, both energy poverty percentage and unemployment rate are below the European average, as well as the households energy costs. Governmental policies created procedures for protecting different categories of vulnerable consumers. In order to inhibit the increased price issues, the government took measures to cap the prices for all population. Policies were also provided for all household renovations. All of these measures led to keeping the energy poverty percentage at a low level.

In Slovakia, we notice that the unemployment rate forecasts are similar to the European average, while the energy poverty percentage is much lower than the same median, mainly due to the fact that household energy costs were somewhat constant over time. The main policies regarding energy poverty involve financial assistance for low income individuals. The general population doesn't benefit from many facilities that allow it to improve household energy efficiency.

Analyzing the collected data for the selected countries, we observed that unemployment rate and energy poverty percentage are interconnected, having comparable tendencies. Using the hybrid model we obtained predictions for unemployment and energy poverty. We conclude that both forecasts are still connected, having similar directions. Unemployment and energy poverty will continue to be strongly correlated, unless governments take measures to diminish the impact that the other aspects have on energy poverty.

## CONCLUSIONS

The countries we considered for this study, Bulgaria, Hungary, Romania and Slovakia, are included in Central and Eastern Europe (*Bouzarovski & Herrero, 2017*) and so they have similar energy-related problems like income inequality, infrastructure issues, buildings that are energetically inefficient, air pollution due to solid fuels usage in households and industry, *etc*.

There is a strong connection between energy price and energy consumption both for industrial energy consumers and households. We observe a significant difference between the household energy consumption in developing countries and developed ones. Multiple developing countries households do not have access to electricity to meet their elemental energy needs and use alternative energy resources like batteries, candles or lamps,

**Table 2   Pearson correlation for UR and EP predicted values significant at the 0.01 level (two-tailed).**

| Country | EP & UR | EP & EU27 EP | UR & EU27 UR |
|---|---|---|---|
| Bulgaria | 0.683 | 0.872 | 0.961 |
| Hungary | 0.856 | 0.791 | 0.768 |
| Romania | 0.773 | 0.946 | 0.912 |
| Slovakia | 0.805 | 0.702 | 0.796 |

that overall are more expensive than electricity. The households not yet coupled to the electricity network could significantly economize if they had a public electricity connection.

Energy poverty implies that individuals cannot keep their house adequately warm or they cannot sustain their basic energy requirements. Based on the many ramifications of energy poverty in the socio-economic climate, minimizing its effects is a priority for European governments.

Unemployment rate has been widely studied by researchers due to its impact on the economic and financial environment both in developing and developed countries, therefore by accurately predicting the unemployment rate, governmental policies could diminish its negative economic influence.

Time series values for unemployment and energy poverty are usually split into two components, a linear and a nonlinear one, due to the complexity of the data and the arbitrary influence of some external factors. Thus, a hybrid model that improves the accuracy of predictions was an appropriate choice, leading us to select the ARIMA + ARNN model for this work.

Based on the generated results and computing the Pearson correlation between the predicted values for unemployment rate and energy poverty levels summarized in Table 2, we can conclude that these two variables have similar trends, both increasing or decreasing at the same rate. In order to lower the energy poverty percentage, the income must be correlated with household energy costs. A low income also generates hidden poverty that could finally lead to energy poverty. Even if we do not take into account the income, the two developing countries are at a higher risk of energy poverty than the EU average.

As shown in Fig. 6, we identified three main issues that influence energy poverty growth: low income, energy efficiency and households energy costs (*European Commission, 2022*). Reviewing the literature, we observed that, in the countries we considered for this study, the connection between these issues is variable at different levels.

Low income is mainly generated by unemployment, low salary, job insecurity or low social protection. Since we estimate that unemployment is the most important factor generating household low income, in this work we computed unemployment predictions in order to analyze its further influence on energy poverty. The income level has repercussions on household energy cost, influenced by energy prices, and on energy efficiency, determined by poor quality of dwellings and appliances, which in turn also generate energy poverty.

We can conclude that households' vulnerability leading to energy poverty is determined by these three causes in different percentages for every country we analyzed. Each of these

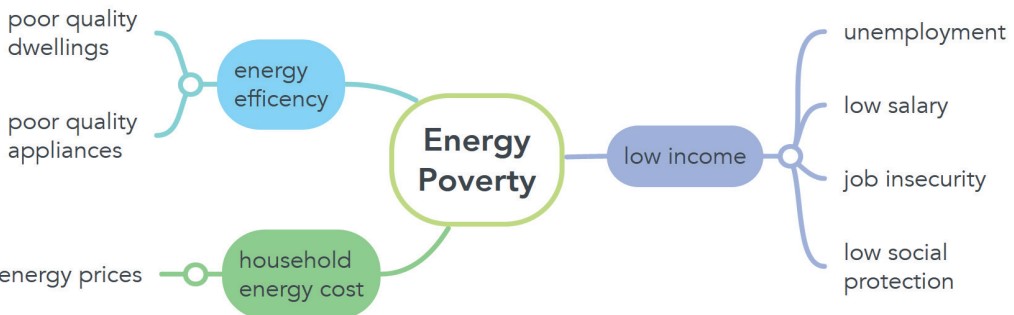

**Figure 6** Main aspects leading to energy poverty.

countries makes great efforts to limit energy poverty effects on households taking into account the specific interconnectivity of these issues.

For some time, the developed countries have adopted measures that facilitate renewable energy installation and usage in households. In the recent period the developing countries have also started to implement programs that provide financial assistance for renewable energy progress.

Limitations of this study include various hidden factors not included in this research but that might also impact the energy poverty levels, factors that could also be considered in a future research, such as the income level, low salaries, job insecurity, or energy prices and household energetic efficiency, in order to provide a better understanding of these findings and their potential relevance in various circumstances. The way other categories of individuals experience energy poverty, like those currently not in the labor market or the retirees, could also be included in further research for a more comprehensive analysis of the topic.

### Funding
The authors received no funding for this work.

### Competing Interests
The authors declare there are no competing interests.

### Author Contributions
- Claudiu Ionut Popirlan conceived and designed the experiments, performed the experiments, analyzed the data, performed the computation work, prepared figures and/or tables, authored or reviewed drafts of the article, and approved the final draft.
- Irina-Valentina Tudor conceived and designed the experiments, performed the experiments, analyzed the data, performed the computation work, prepared figures and/or tables, authored or reviewed drafts of the article, and approved the final draft.

- Cristina Popirlan conceived and designed the experiments, performed the experiments, analyzed the data, performed the computation work, prepared figures and/or tables, authored or reviewed drafts of the article, and approved the final draft.

## Data Availability

The data is available at the World Bank: https://data.worldbank.org/indicator/SL.UEM.TOTL.ZS

Unemployment, total (% of total labor force) (modeled ILO estimate) International Labour Organization. "ILO Modelled Estimates and Projections database (ILOEST)" ILOSTAT. Accessed February 21, 2023. https://ilostat.ilo.org/data/. (CC BY-4.0).

The data is also available at EUROSTAT:

https://ec.europa.eu/eurostat/databrowser/view/SDG_07_60/default/table?lang=en{&}category=sdg.sdg_07. Population unable to keep home adequately warm by poverty status. online data code: SDG_07_60. Last update: 09/03/2023.

## Supplemental Information

Supplemental information for this article can be found online at http://dx.doi.org/10.7717/peerj-cs.1464#supplemental-information.

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
