# Peer review of "Predicting the unemployment rate and energy poverty levels in selected European Union countries using an ARIMA-ARNN model"

_PeerJ Computer Science, doi:10.7717/peerj-cs.1464_

## Round 0.1 · original submission · Major Revisions

Please revise your paper according to the reviewers' comments.
Thanks.

Reviewer 1 ·

Basic reporting

Overall is well written using good English style

Experimental design

The ARIMA-ARNN model is briefly explained

Validity of the findings

The title of the manuscript is "Predicting the effect of unemployment on energy poverty levels in selected European Union countries using an ARIMA-ARNN model". The method used is the ARIMA-ARNN model to predict energy poverty and unemployment rates in two developing countries, Romania and Bulgaria, and in two developed countries, Hungary and Slovakia, using European datasets.

In line 87-90, the authors stated that ARIMA model is suitable for linear forecasting while ARNNs are more convenient for nonlinear predictions and they used both models into a combined hybrid model to obtain the optimal predictions.

My question: what is the justification of the statements in line 359-362, line 368-369, line 388-386 that stated there is strong positive relationship between unemployment rate and energy poverty?
The trends presented in Figures 4 and 5 cannot show the magnitude and direction of the relationship between unemployment rate and energy poverty.
The author need to provide a measurement to justified that there is a relationship between unemployment rate and energy poverty.
In addition, the author should add references to the framework presented in Figure 6.

Additional comments

My suggestion is the authors change the title of the manuscript to be in line with the method used in their study (i.e., to predict) and remove all discussion related to "relationship" and "effect".

Reviewer 2 ·

Basic reporting

This manuscript explores the correlation between energy poverty and unemployment rates in the context of four East European countries, both developed and developing. The study employs a model that considers both linear and non-linear factors, and is supported by a comprehensive dataset and code.

The authors may benefit from reorganizing their manuscript for greater clarity. For instance, combining the introduction and literature review sections, and placing the data analysis in the results section may provide a clearer outline. The current arrangement of combining existing literature with new analysis within a single section can be confusing for readers.

The study's figures, namely Figure 1 and Figure 2, utilize different time windows to present the data. While Figure 1 displays a trend with a six-month frequency, Figure 2 presents daily data. For consistency and ease of comparison, it would be beneficial to standardize the temporal granularity and model used for presenting both figures.

Experimental design

The authors have noted that a previous study had explored a similar topic, and that the present study primarily expanded upon the previous study by examining additional years (lines 68-80). However, it remains unclear whether the present study has uncovered any new findings beyond reaffirming the earlier study's conclusions.

The study's objective is not clearly defined, as it appears to address both inference (lines 62-63) and forecast (line 92) using the same dataset. The authors' emphasis on these two types of analyses appears imbalanced, as the "literature review and data analysis" section focuses on establishing the correlation between energy poverty and unemployment rates, while the "methods and results" section primarily focuses on forecasting.

There is a minor issue with the authors' definition of developed and developing countries. For example, while Romania is not as developed as Hungary and Slovakia, some source considers it to be a developed country based on the United Nations' Human Development Index (https://www.romania-insider.com/romania-retains-developed-country-status-un-ranking#:~:text=Romania%20kept%20its%20status%20as,globally%20from%20among%20EU%20countries.). The authors should clarify their definition of these terms to ensure consistency and accuracy.

The World Bank defines unemployment rate as the proportion of the labor force that is available for and seeking employment but currently without work. However, this study does not address how individuals not currently in the labor market, such as retirees, experience energy poverty. It would be beneficial for the authors to consider these individuals' situations to provide a more comprehensive understanding of the study's topic.

Validity of the findings

As the authors have noted in their observations on Bulgaria (lines 333-334, 344-346), income appears to be a more direct indicator of energy poverty. However, it remains unclear why the authors did not consider other hidden factors, such as low salaries and job insecurity, in their analysis of energy poverty.

Additionally, it is important for the study to address its limitations, such as hidden factors that were not considered and the robustness of its conclusions when applied the conclusion to other societies. Discussing these limitations would help to provide a more complete understanding of the study's findings and their potential applicability in different contexts.

---

## Round 0.2 · accepted · Accept

The authors have issued most concerns of reviewers. Please address the formatting issues in production.

Reviewer 1 ·

Basic reporting

No comment

Experimental design

No comment

Validity of the findings

No comment

Additional comments

No comment

Reviewer 2 ·

Basic reporting

The updated version needs to readjust the heading style. For instance, "INTRODUCTION" and "1 METHODS AND RESULTS" should be at the same level, but one without number, the other has the leading number "1." Unless there is an explicit requirement from the journal, a clearer structure could be 1. introduction; 2. Methods; 3. Results; 4. Discussion; and 5. Conclusions.

Experimental design

The revised version looks good.

Validity of the findings

I am satisfied with the revised version.